# Characterization of the Peri-Membrane Fluorescence Phenomenon Allowing the Detection of Urothelial Tumor Cells in Urine

**DOI:** 10.3390/cancers14092171

**Published:** 2022-04-26

**Authors:** Charly Gutierrez, Xavier Pinson, Kathleen Jarnouen, Marine Charpentier, Raphael Pineau, Laëtitia Lallement, Rémy Pedeux

**Affiliations:** 1University Rennes, INSERM, OSS-UMR_S 1242, CLCC Eugène Marquis, 35042 Rennes, France; charly.gutierrez@univ-rennes1.fr (C.G.); m.charpentier75@laposte.net (M.C.); raphael.pineau.1@univ-rennes1.fr (R.P.); 2University Rennes, CNRS, Inserm, Biosit UAR3480 US_S 018, MRic Core Facility, 35000 Rennes, France; xavier.pinson@univ-rennes1.fr; 3VitaDX International, 74F Rue de Paris, 35000 Rennes, France; kjarnouen@cell4tech.fr (K.J.); laetitia@vitadx.com (L.L.)

**Keywords:** bladder cancer, fluorescence, diagnostic test, urine, urinary cytology, peri-membrane fluorescence, liquid biopsy, non-invasive

## Abstract

**Simple Summary:**

To detect bladder cancer (BC), urinary cytology and cystoscopy are the primary diagnostic tests used. Urine cytology is non-invasive, easy to collect, with medium sensitivity and high specificity. It is an effective way to detect high-grade BC, but it is less effective on low-grade BC because the rate of equivocal results is much higher, making them difficult to detect. Despite the implementation of new diagnostics, urinary cytology and cystoscopy remain the gold standard. Instead of looking for new diagnostics, one of the new research areas is the improvement of urinary cytology. Recently, the fluorescent properties of plasma membranes of urothelial tumor cells, called peri-membrane fluorescence, found in urinary cytology have been shown to be useful in improving the early detection of BC. The main objective of this study was to characterize the peri-membrane fluorescence allowing the detection of urothelial tumor cells in urine.

**Abstract:**

Urine cytology is non-invasive, easy to collect, with medium sensitivity and a high specificity. It is an effective way to detect high-grade bladder cancer (BC), but it is less effective on low-grade BC because the rate of equivocal results is much higher. Recently, the fluorescent properties of plasma membranes of urothelial tumor cells (UTC) found in urine cytology have been shown to be useful in improving the early detection of BC. This phenomenon is called peri-membrane fluorescence (PMF). Based on previous studies that have identified the PMF on UTCs, the main objective was to characterize this phenomenon. For this study, a software was specially created to quantify the PMF of all tested cells and different treatments performed. PMF was not found to be a morphological and discriminating feature of UTCs, all cells in shape and not from urine show PMF. We were able to highlight the crucial role of plasma membrane integrity in the maintenance of PMF. Finally, it was found that the induction of a strong cellular stress induced a decrease in PMF, mimicking what was observed in non-tumor cells collected from urine. These results suggest that PMF is found in cells able to resist this stress, such as tumor cells.

## 1. Introduction

Bladder cancer (BC) accounts for 90% of urothelial cancers. It is the 10th most common cancer worldwide, with an estimated 572,000 new cases and 212,000 deaths in 2020 [1], and ranks second only to prostate cancer, among urological cancers. The countries with the highest rates of BC are in North America, Southern, and Western Europe. At least 90% of cases are urothelial carcinomas or transitional cell carcinomas (TCC). They are classified according to the TNM system [2]. These carcinomas are divided into two categories, non-muscle-invasive bladder cancer (NMIBC) and muscle-invasive bladder cancer (MIBC). NMIBCs account for approximately 70–85% of diagnosed TCCs [3]. NMIBCs are an early stage of cancer and are classified as Ta to T1 and are found in the first layer (urothelium) and second layer (lamina propria) of the bladder. MIBCs are classified as T2 to T4: T2 is found in the third layer of the bladder (muscles), T3 in the fourth layer (perivesical tissue), and T4 invades adjacent tissue. Estimated survival for NMIBC is greater than 78% at 5 years, whereas for MIBC, a more aggressive tumor, it is estimated to be less than 50%. Therefore, early detection of TCC is important [4].

To detect BC, urine cytology and cystoscopy are the primary diagnostic tests used. Urinary cytology is non-invasive, easy to collect, and with an overall sensitivity of 38% and a specificity of 98% [5]. It is an effective way to detect high-grade BC (improved sensitivity between 52 and 78%), but it is less effective on low grades than high grades because the rate of equivocal results is much higher, making them difficult to detect [5]. Because of the overall low sensitivity of the urine cytology, new diagnostic tests using urine, and whose sensitivity is superior to that of urine cytology are being sought. It is also important that these diagnostic tests easily detect BC at an early stage and facilitate the follow-up of patients. This is why the search for urinary biomarkers of BC has been developed in recent years, such as circulating cell-free DNA, microRNA, circular RNA, non-coding RNA, proteins, cells, etc. [3,6,7,8]. Since the beginning of the implementation of urine cytology, only three new diagnostic methods based on urine BC protein markers have been validated by the Food and Drug Administration (FDA). BTA-TRAK™/STAT™ which quantifies the bladder tumor antigen [9,10], NMP22 (nuclear matrix protein 22) protein test which quantifies the NMP22 protein [11,12], and ImmunoCyt™/uCyt+™ which targets two antigens (the carcinoembryonic antigen [13] and mucinous bladder antigen [14]). Three other methods have also been validated by the FDA and focus more on searching for DNA or RNA released or contained by malignant cells in the urine: UroVysion™ which is a multitarget fluorescence in situ hybridization that detects aneuploidy [15], Xpert a bladder cancer monitor which analyzes five mRNA targets frequently overexpressed in BC [16,17], and CxBladder detect which quantifies five mRNAs associated with the growth and propagation of tumor tissue and with inflammation. One of the advantages of using DNA or RNA is that it can be amplified and can be used at an early stage. Despite the introduction of these new diagnostics, they are, at best, used in combination with urine cytology or only for the follow-up of patients [18]. Urine cytology remains, along with cystoscopy, the diagnostic test of choice because of its higher specificity, although sensitivity may sometimes be lower than other diagnostics [19]. Some diagnostic tests also require specific skills on the part of pathologists, or the preparation of additional samples, resulting in limited access to these tests, while urine cytology remains the simplest diagnostic test to perform [20].

That is why, instead of developing new diagnostics, one of the new research directions is the improvement of urine cytology. Steenkeste et al. showed fluorescent properties of Papanicolaou-stained urine cytology after excitation at 488 nm [21]. These observations revealed the presence of different fluorescence patterns on urothelial cells (UC) under various clinical conditions without any intervention in the collection, fixation, and staining protocol. It was observed on these fluorescent cytologies that normal and quiescent UC are characterized by a homogeneous and uniform fluorescence on cells, while cells from BC showed a higher fluorescence at the plasma membrane than in the cytoplasm. This difference in fluorescence between the plasma membrane and the cytoplasm of BC cells is called peri-membrane fluorescence (PMF). This same research group has therefore proposed to use these fluorescence patterns to differentiate between healthy and tumor cells and thus facilitate the detection of BC. This method could be implemented using a single urine sample, without the need to modify the Papanicolaou staining protocol, unlike other methods using fluorescence as a photodynamic diagnostic [22,23]. This PMF phenomenon is still poorly understood. Understanding the reaction induced to differentiate urothelial tumor cells (UTC) from healthy UCs is essential for the development of a new diagnostic method.

The main objective of this study was to characterize the PMF phenomenon allowing the detection of UTC in the urine. In this study, it has been shown that the morphological characteristics of the cell were not responsible for the PMF and, furthermore, PMF was found on all the cells of various origins. It was hypothesized that a physiological change in the healthy cell was induced during bladder cell detachment. First, a loss of PMF was observed when cell membranes were disrupted by permeabilization. Second, we have shown that induction of senescence, one of the cellular mechanisms involved in bladder cell detachment [24,25], was not sufficient to induce the loss of the PMF, a more aggressive stress was required. Thus, our results suggest that the presence of a PMF is due to the ability of tumor cells to survive the stressful conditions of their environment, unlike healthy cells.

## 2. Materials and Methods

### 2.1. Cytological Slides

Urinary cytological slides coming from patients were given by VitaDX and were prepared according to the protocol explained by Steenkeste et al. [21] and adapted by VitaDX according to their patent [26]. Urine was gathered in a sterile container. The sample was concentrated by centrifugation in 50 mL Falcon vials (Becton Dickinson, Franklin Lakes, NJ, USA) for 10 min at 600× *g* at room temperature. The supernatant was removed by inversion. The pellet was pipetted into PreservCyt^®^ collection vials (Hologic, Marlborough, MA, USA) and incubated for 15 min at room temperature. Cells transferred on slides was performed with a ThinPrep 2000 Autoloader System (Hologic) and then were stained using the Papanicolaou technique [27] with a Dako CoverStainer automate CS1000 (Agilent, Santa Clara, CA, USA). Slides were covered by a Tissue-Tek^®^ Film Automated Coverslipper (Sakura, Japan) with Tissue-Tek^®^ Coverslipping Film (Sakura, Japan) using the VitaDX protocol at FOCH Hospital. A minimum of 12 h of drying time was required before scanning the slides. The slides were scanned in both transmission and fluorescence using a Hamamatsu Nanozoomer-S60 slide scanner with its additional fluorescence module (×40, Inc. Hamamatsu, Shizuoka, Japan).

### 2.2. Cell Line Culture

Human urinary bladder carcinoma T-24 cell line, human lung carcinoma A549 cell line, human non-small cell lung cancer H1975 cell line, human osteosarcoma U2OS cell line, human lung carcinoma H69 cell line, lung epithelial hTERT-immortalized HBEC3-KT cell line, and lung adenocarcinoma HCC827 were purchased from the American Type Culture Collection (ATCC) (Manassas, VA, USA). Human lung adenocarcinoma PC-9 was purchased from Sigma-Aldrich (St. Louis, MO, USA). Immortalized embryonic human fibroblasts MRC5-hTERT were generated by the Curtis Harris Laboratory [28]. Primary epithelial cells derived from normal human bladder HBlEpC were purchased from Cell Applications (Inc., San Diego, CA, USA). Cell lines were maintained in 5 mL (25 cm^2^), 15 mL (75 cm^2^), or 20 mL (150 cm^2^) Falcon flasks (Becton Dickinson, Franklin Lakes, NJ, USA). T24 and U2OS cell lines were cultured in McCoy’s 5A medium (Gibco; Thermo Fisher Scientific, Inc., Waltham, MA, USA). A549 and MRC5-hTERT cell lines were cultured in DMEM. H1975 and PC-9 cell lines were cultured in RPMI-1640 growth medium (Gibco; Thermo Fisher Scientific, Inc., Waltham, MA, USA). HBEC3-KT cell lines were cultured in DMEM/F12 (Gibco; Thermo Fisher Scientific, Inc., Waltham, MA, USA). HBlEpC were cultured in bladder epithelial cell growth medium (Cell Applications, Inc., San Diego, CA, USA), according to the manufacturer’s instruction. All cells were supplemented with 10% fetal bovine serum, except HBlEpC, and cultured at 37 °C in a humidified 5% CO_2_ incubator.

### 2.3. Cytological Slides with Cell Line

In order to work similar to cytological slides obtained from patients, cells were trypsinized with 0.05% trypsin (Gibco; Thermo Fisher Scientific, Inc., Waltham, MA, USA) and counted on Malassez cells. For one slide, 80,000 cells were recovered and then fixed in PreservCyt^®^. Cells transferred on slides was performed with a ThinPrep 5000 Autoloader System (Hologic) and then were stained using the Papanicolaou technique with a Dako CoverStainer automate CS1000. Slides were covered by a Tissue-Tek^®^ Film Automated Film Coverslipper with Tissue-Tek^®^ Coverslipping Film. After 24 h of drying, slides were acquired by a NanoZoomer-S60 Digital slide scanner in transmission mode (bright field light) and in fluorescent mode (488 nm line of argon lasers) with a 524/24, 485/20 nm BrightLine^®^ single-band bandpass filter (Semrock, Rochester, NY, USA), or 506 nm edge BrightLine^®^ single-edge standard epi-fluorescence dichroic beamsplitt (Semrock, Rochester, NY, USA). Slides were also acquired with a confocal SP8 T. C. S. Leica microscope (×40, Confocal Scan Head Leica SP8, Wetzlar, Germany) in fluorescent mode (488 nm line of argon lasers), according to the protocol explained by Steenkeste et al. [21]. Images acquired with the NanoZoomer were analyzed with NDP.view2 software [29] and images acquired with the SP8 confocal were analyzed with Fiji software [30].

### 2.4. Phalloidin Staining 

The T24 cell line was cultured as previously described. Cells were trypsinized and counted on Malassez cells. T24 cells were washed three times with warm PBS (Gibco; Thermo Fisher Scientific, Inc., Waltham, MA, USA). Then, a maximum volume of 5 mL containing 80,000 cells was collected and then cytospinned on a slide for 10 min at 800 g at RT. Cells were then fixed with 4% paraformaldehyde for 15 min and washed three times with PBS. T24 cells were then stained, according to the manufacturer’s instruction, with Phalloidin-iFluor 488 (Abcam plc. Cambridge, UK) and Hoechst (33,258, Sigma-Aldrich, St. Louis, MO, USA). The slide was sealed with Mowiol^®^ 4–88 mounting media (81,381, Sigma-Aldrich) and a coverslip. Images were acquired with a confocal SP8 TCS Leica microscope.

### 2.5. Quantification Software

Perifluo QCG [31] is a software that has been specially created to determine the ratio of the PMF to cytoplasm fluorescence while deducting the background from a urine cytology or a cell line cytology that was acquired with the NanoZoomer in transmission and fluorescent mode (Appendix A). The cytology that was acquired with the Nanozoomer was saved in an “.ndpis” format. In this format, the transmission and fluorescent parts were merged into a single image. The conversion of the “.ndpis” format into the “.tif” format and the separation of the transmission mode from the fluorescent mode into two distinct images were mandatory steps for the continuation of the software. To separate fluorescence from transmission, the image was exported in “.tif” format with the NDP.view2 software (Hamamatsu) where it was possible to choose the channel of the image to export. The image was zoomed to the ×0.9 objective to obtain a visual of the whole cytology and recorded digitally magnified 10 times. The fluorescence image was renamed “fluo.tif” and the transmission image “bf.tif”. Both images were saved in the same folder so that the Perifluo QCG software could find them. Perifluo QCG is composed of two sub-software developed under Fiji. The first one merges the “fluo.tif” and “bf.tif” images into a single “.tif” image with both channels. It was then required for the operator to delimit a square of 12,000/12,000 pixels that encompasses the cytology. With this square, the software can divide the merged image into 16 sub-images of 3000/3000 pixels in order to simplify the software’s calculation load. Then, they were saved in the same folder where the “bf.tif” and “fluo.tif” images were already saved. The second part of the software searches and determines the surface of each cell with the help of the ROI, image by image. This step was made on the “bf.tif” part of the image. The ROI search allows the software to determine the size of the cell, where the edge of the cell and where the cytoplasm is. Only cells with a diameter between 1000 and 10,000 pixels were retained. Once cells were located and analyzed, the data were reported on the “fluo.tif” image. After the data were transferred, the software could define the edge of each cell according to their previously determined ROI. Then, the software determined a perimeter of 5 pixels around the cell to calculate the PMF. At the end, an open torus was obtained. This geometrical figure encircles the cytoplasm and the cylinder of the torus represents the PMF. The general intensity of the cytoplasm and the PMF were then calculated by the software and the result was reported in a spreadsheet with the cell number. The fluorescence ratio (RF) of PMF of each cell was calculated in Excel (Microsoft) with the formula:$$FR\;of\;one\;cell=\frac{intensity\;of\;plasma\;membrane\;fluorescence\;\left(or\;PMF\right)}{intensity\;of\;cytoplasm\;fluorescence}$$

### 2.6. Animals

Rat Sprague Dawley males, 300 g (Janvier, Laval, France), were used. All animals were treated in accordance with the European Community Directive guidelines (Agreement B35-238-40 Biosit Rennes, France/No DIR 5569) and were approved by the local ethics committee, ensuring the breeding and daily monitoring of the animals in the best conditions of well-being according to the law and the 3R rule (reduce, refine, and replace). The rats were housed in a specific pathogen-free level laboratory under controlled conditions (12 h light/dark cycle, controlled temperature, and humidity, ad libitum access to food and water).

### 2.7. Experimental Protocol

All rats used in this study had to be sacrificed for research not associated with this study. For the AM conditions, rats were sacrificed in the morning, and for the PM conditions, in the afternoon. Three to 5 rats were pooled by sample. Urine was collected by suprapubic puncture (±1 mL/rat) immediately after dissection of the rats and was then kept in PreservCyt^®^ before staining protocol. After puncture, bladders were collected and maintained at 37 °C in PBS until they were used. To isolate UC from bladders, we followed the protocol of T. Kloskowski et al. [32]. Once bladders were collected, they were turned upside down and placed in a 15 mL Falcon (Becton Dickinson, Franklin Lakes, NJ, USA) tube containing a 0.05% trypsin solution. Falcon tubes were gently shaken for 2 h at 37 °C. Trypsin was then inactivated by the addition of a serum-supplemented culture medium and the bladders were removed and opened to be laid flat. Once they were flattened, they were gently scraped in warm PBS to obtain as many cells as possible. The cells obtained after scraping were put back into the Falcon containing the culture medium and trypsin. The tube was then centrifuged at 400 G for 5 min. The supernatant was removed and the cells were resuspended in 1–2 mL of culture medium before being placed in the PreservCyt^®^. After fixation, urine and bladders were stained according to the protocol described by Steenkeste et al. [21].

### 2.8. Permeabilization Assays 

After trypsinization, T24 cells were counted in Malassez cells. 80,000 T24 cells were recovered and transferred to a 1.5 mL Eppendorf tube (Sigma-Aldrich) containing PBS for the condition without Triton or Triton 0.4% for the condition with Triton. For each condition, one volume of cells was added to one volume of PBS or Triton 0.4% to obtain a final concentration of 0.2%. Cells were incubated for 90 s under agitation. Approximately 300 mL of urine was collected in 50 mL Falcons from healthy volunteers. The urine was centrifuged at 700 G for 4 min. The supernatant was discarded. The urine was then resuspended in 3 mL of PBS. For the Urine condition, 1 mL of urine was directly put into a jar of fixative. For the Urine + T24 condition, 1 mL of urine and 80,000 T24 cells were put into PreservCyt^®^. For the Urine + P.T24 condition, 1 mL of urine and 80,000 T24 cells permeabilized, as described above, were put into a fixative jar. All three conditions were then stained according to the protocol described by Steenkeste et al. [21].

### 2.9. Induction of Cellular Stress

Cells were seeded at a density of 40,000 cells/cm² 24 h before the treatment. For senescence (Irradiated/Untreated condition), cells were only irradiated with X-ray (20 Gy, CellRad, Faxitron [no filter] 130 kV, 5 mA). For the induced cell stress condition (Irradiated/Treated condition), and according to the protocol explained by Correia-Melo et al. [33], cells were irradiated with 20 Gy and treated with neocarzinostatin (80 ng/mL) and H_2_O_2_ (400 µM) in serum-free media for 1 h. Following treatment, culture medium was refreshed with serum-free media. Etoposide treatment (50 µM) was performed continuously every 3 days for 9 days for Irradiated/Treated condition. Cells were cultured with serum-free media for 9 days. Then, cells were recovered and put in PreservCyt^®^ for the staining or washed with PBS 3 times for cell death assay.

### 2.10. Western Blot

Whole-cell protein extracts were prepared for immunoblotting by cell lysis with RIPA buffer (Cell Signaling; Danvers, MA, USA) in combination with a protease inhibitor cocktail (Cell Signaling). Protein samples were subjected to electrophoresis using the NuPAGE 12% Bis-Tris Gels Electrophoresis system (Invitrogen, Carlsbad, CA, USA). The antibodies used in this study were β-actin (Sigma-Aldrich), P21 (Santa Cruz, Dallas, TX, USA), anti-mouse and anti-rabbit IgG, and HRP-linked antibody (Cell Signaling Technology). 

### 2.11. Cells Death Assay and Flow Cytometry

For cells death assay, cells were cultured as described in “Induction of cellular stress” and stopped with trypsinization after 1 h of treatment with etoposide for J + 0, after 3 days of treatment for J + 3, after 6 days of treatment for J + 6, and after 9 days of treatment for J + 9. Cells were washed 3 times with cold PBS and were centrifuged at 700 G for 4 min. Annexin V staining was performed with FITC Annexin V Apoptosis Detection Kit I (BD biosciences, Inc., Franklin Lakes, NJ, USA) according to the manufacturer’s instructions. Cells were then acquired on a Gallios flow cytometer (Beckman Coulter, Brea, CA, USA) and the data were analyzed using Kaluza software. Total cell death was determined by summing the percentage of cells positive for annexin V with a percentage of cells positive for propidium iodide and with a percentage of cells positive for both markers. 

### 2.12. Statistical Analysis

Statistical analyses of PMF quantification were performed with one-way analysis of variance (ANOVA) to calculate significant differences between groups using Prism 8 (8.4.3). A *p*-value of <0.05 was considered statistically significant.

## 3. Results

### 3.1. Role of the Nucleus and the Cytoplasm in the PMF Phenomenon

The analysis of Papanicolaou-stained urine cytology is usually performed only in bright field light; only very recently has the utility of analysis in transmitted light been demonstrated [21,34]. The use of fluorescence on Papanicolaou-stained UC facilitates the visualization of different states of the UC on the urine cytology of a patient. A healthy UC will fluoresce evenly throughout the cell (Figure 1A and Appendix A). In advance and early-stage UTC, fluorescence will be much more prominent in the plasma membrane (periphery) than in the cytoplasm and nucleus (Figure 1A,B and Appendix A). The presence of such fluorescence on the UTC is referred to as peri-membrane fluorescence (PMF) and allows for an improved and easier diagnosis of the BC. The main characteristic that allows the diagnosis of a UTC with bright field light is the presence of a nucleus that occupies at least 70% of the cell space [35]. It is possible to find UTC in which the nucleus takes up almost all of the cytoplasm, which is why we were interested in determining whether the size of the nucleus plays a role in the PMF phenomenon and therefore whether it is due to the morphological features of the UTC. Thus, we investigated whether the presence of a large nucleus would take up the entire cell volume and engulf almost the entire cytoplasm. Only a small part of the cytoplasm, located close to the plasma membrane, would then be visible and thus responsible for the PMF phenomenon (Figure 1A,B and Appendix A), similar to the moon covering the sun during an eclipse. We speak then of an eclipse effect where the nucleus plays the role of the moon and the cytoplasm of the sun. The T24 cell line was used to study the BC. This cell line was derived from a malignant bladder cancer undergoing progression to the muscle layer of the bladder wall. To determine if the nucleus was able to induce this effect, we labeled the cytoplasm with phalloidin and the nucleus with DAPI. Following the labeling, we found that the nucleus did not take all the space in the cell (Figure 1C) and therefore did not induce this eclipse effect. We conclude that the PMF is not due to this effect. In summary, we have shown that the size of the nucleus does not intervene in the PMF phenomenon and that it is indeed a peri-membrane phenomenon.

### 3.2. PMF Expressed on Cells Line from Various Origins

As described above, the cells which express a PMF in patient urine cytology are generally UTCs. Our BC model, the T24 cell, has the same PMF (Figure 1B and Appendix A) as the UTC found in BC patients and is therefore valid for the study of PMF. For comparison purposes, we wanted to know if other cell lines were able to express this PMF phenomenon. For this purpose, we first wanted to determine if only bladder carcinoma cells expressed this PMF by comparing them to other non-bladder epithelial carcinomas cell lines. For this first comparison, we used cells in suspension after detaching them with trypsin in order to mimic what is found in urine with the UTC (Figure 2A). As can be seen, whatever the cell derived from non-bladder epithelial carcinomas, we observe the presence of a PMF identical to T24 cells. We therefore wanted to investigate whether this PMF was only present on the cell lines from non-bladder epithelial carcinomas by comparing them with the non-epithelial carcinomas cell lines. For this purpose, we used a line derived from the immortalized human embryonic fibroblasts (MRC-5 hTERT), a malignant tumor of bone origin (U2-OS), and a small cell lung carcinoma (H69) (Figure 2C). Like the previous cell lines, PMF was also found in cell lines not derived from epithelial carcinomas. Based on the fact that UCs are epithelial cells and do not express any PMF on urine cytology, it was hypothesized that only the normal healthy epithelial cells do not express PMF. To validate this hypothesis, a normal human bronchial epithelial line immortalized with CDK4 was tested. However, in this case, a PMF was also found (Figure 2E). Finally, we used a primary bladder cell line to determine if only healthy bladder cells did not express PMF, but even with this line, PMF is observed (Figure 2F and Appendix A). We then wanted to determine whether protein mechanisms involved in cell adhesion played a role in the PMF phenomenon. For this, we used adherent cell lines from non-bladder epithelial carcinomas and cell lines not from epithelial carcinomas. The cells were grown directly on slides and then stained with Papanicolaou without being trypsinized (Figure 2B,D). It can be seen that in this case, the fluorescence is always more pronounced at the plasma membrane than in the cytoplasm. Whether the cell is adherent or in suspension, a PMF is always observed. Thus, cell adhesion is not involved in this phenomenon. We therefore show here that cells, whether healthy or tumor cells, in suspension or adherent, exhibit PMF. To validate whether all cells exhibit identical PMF or cell type-dependent PMF, it was important to examine the fluorescence in more detail and move from a qualitative to a quantitative analysis. The advantage of a quantitative analysis is that if there is a variation in fluorescence between different cell lines that express a PMF, it will be possible to distinguish it. This allows to determine which type of PMF the cells express according to their physiological state or tissue origin. To determine the PMF in a quantitative way, a software capable of determining precisely the membrane and cytoplasmic fluorescence was developed. This software, Perifluo QCG [31], is able to detect every cell present in the sample. When a cell is detected, its cytoplasmic and peripheral/membrane fluorescence is quantified while deducting the background. At the end of the process, the PMF and cytoplasmic fluorescence of each cell have been quantified. A fluorescence ratio (FR) is then determined on all cells by using the previously obtained data.

This software allows us to determine a PMF threshold where all samples with an average FR greater than 1 were considered samples with a majority of cells expressing a PMF and conversely for samples with an average FR less than 1. After quantification, it was found that the FR of the different cell lines was higher than 1 and there was no significant difference between the different cell lines analyzed (Figure 2G and Appendix A). The presence of an FR higher than 1 on all the cells analyzed confirmed that they all express a PMF. None of the cells tested (Figure 2H) were detected without PMF. Moreover, the absence of a significant difference between the PMF of the tested cell lines confirmed that the PMF is cell type dependent but is found in all cells in all cases. We can therefore conclude that regardless of the type of cell line used and its status, the PMF phenomenon was always observed. Based on the fact that before being found in urine, UCs are present in their tissue of origin, the urothelium that lines the bladder [36], and all the cells tested showed a PMF, it was hypothesized that the passage of UCs from the urothelium to the urine induced a change in PMF which cannot be seen in cell culture.

### 3.3. Urothelial Cells Collected in the Urine versus the Urothelial Cells Collected in the Urothelium

In order to understand the differences in PMF of the UCs recovered in the urine (UCU), the healthy rat model was used to determine whether or not UCUs recovered by suprapubic puncture express a different PMF than UCs recovered from the UroThelium (UCUT). The rat model was chosen because the quantity of urine and the size of the bladder are much larger than in the mouse. For each condition, between 2 and 5 rats were analyzed. It was found, by quantitative analysis, that the general FR of UCUs is less than 1 (ranging from 0.62 for the minimum to 0.85 for the maximum) (Figure 3A and Appendix A), confirming that the UCUs do not show a PMF. In contrast to urine, for UCUTs we observe a clear presence of PMF (Figure 3B and Appendix A). Thus, there was a very significant increase in the FR for UCUTs compared to UCUs, with an FR of 1.081 and 0.708, respectively (Figure 3C). It can therefore be concluded that UCs do not have the same PMF when recovered from urine or directly from urothelium.

### 3.4. Role of the Plasma Membrane Integrity on the PMF

With these results, we hypothesized that the shift in PMF is due to a cellular modification induced during the transition from the urothelium to the urine. We thought that this modification of the PMF was possibly induced during the normal detachment of UCs from the urothelium towards the urine. Detachment of UCs is frequently caused by more or less aggressive cellular stresses such as senescence, apoptosis, or injuries [36,37]. Furthermore, once the cells are suspended in the urine, the lack of cell–cell interaction leads to the activation of anoikis, a phenomenon responsible for the entry of the cells into apoptosis [38,39]. All of these phenomena often lead to a metabolic change in the cell which can result in a modification of its composition or its integrity, in particular at the level of the plasma membrane which can become permeable or change its composition [40,41,42]. To understand the importance of plasma membrane integrity, a permeabilization of the plasma membrane was induced in T24 cells to mimic this phenomenon potentially undergone by UCUs during stress and to determine its involvement in PMF. Triton was used to recreate the phenomenon of permeabilization. Triton is a monomeric detergent able to insert itself into the lipid membrane, ultimately permeabilizing the membranes. To determine the importance of membrane integrity on the PMF, T24 cells were treated with 0.2% Triton-PBS for 90 s. After treatment, a visual decrease in PMF was observed (Figure 4A). Following these results, we wanted to know if it was possible to recreate what was observed with UCUs. To do so, we compared the FR of urine from a healthy patient versus urine from a healthy patient supplemented with T24 cells versus urine from a healthy patient supplemented with T24 cells permeabilized with 0.2% Triton (Figure 4B and Appendix A). It was found that the FR of the urine supplemented with permeabilized T24 cells was less than 1 and did not differ significantly from the urine of the healthy patient (0.9022 and 0.9489, respectively) (Figure 4B and Appendix A). On the contrary, the FR of urine containing non-permeabilized T24 cells was found to be greater than 1 (1.16) and was very significantly different from both healthy and permeabilized T24 cell-containing patient urine (Figure 4B and Appendix A). Therefore, we were able to recreate what is observed with UCUs and the integrity of the membrane is very important for the PMF phenomenon since its disruption leads to a loss of the PMF. 

### 3.5. Influence of the Cellular Stresses on the PMF

After having observed that the integrity of the membrane is very important for the PMF, we wanted to know if instead of inducing an alteration of the plasma membrane using detergent it could be induced by the activation of cellular stresses. Activating cellular stresses would bring us closer to the physiological phenomenon observed during the passage from the urothelium to the urine and thus recreate what is found in the UCU. Thus, immortalized healthy MRC-5-hTERT cells were first studied. Senescence was induced, as described by Correia-Melo et al. [33] by irradiating them with X-rays (20 Gy). This senescence condition was called the Irradiated/Untreated condition (I/U). To induce a stronger cellular stress than senescence, cells were treated according to the protocol of Correia-Melo et al. [33]. An initial X-ray treatment was performed (20 Gy) followed by a one-hour incubation in fresh serum-free medium containing neocarzinostatin (NCS) and H_2_O_2_. Then, the medium was removed and replaced with fresh serum-free medium supplemented with etoposide. This cellular stress condition was called the Irradiated/Treated condition (I/T). After nine days of treatment, a significant decrease in the FR of MRC-5-hTERT was observed for I/T conditions compared with the Unirradiated/Untreated (U/U) and I/U conditions (Figure 5A,B and Appendix A). To confirm the involvement of the stress in the decrease/loss of the PMF, we wanted to know if it was possible to obtain these same observations with our bladder cancer model whose cells are known to always express a PMF. For this purpose, cells were cultured under the same conditions as MRC-5-hTERT. After treatment, I/T T24 cells were also found to express significantly lower FR than U/U and I/U T24 cells (0.761 vs 0.951 and 1.1124, respectively) (Figure 5A,C and Appendix A). To confirm the induction of senescence, the activity of P21, a major protein involved in senescence [43], was determined. Western blot analysis revealed that P21 was found more expressed in the I/U condition than in the U/U condition (Figure 5D), confirming the activation of senescence by X-ray irradiation. Senescence activation alone in the I/U condition is not sufficient to induce the decrease in PMF, in contrast to the I/T condition where it was required to induce a strong cellular stress in addition to senescence activation (Figure 5D). To confirm the importance of the treatment, it was decided to separate the different phases and molecules used to induce a cellular stress (Figure 5E,F and Appendix A). The use of etoposide was determined to be the major and determining factor in the decrease in PMF. A synergistic effect was observed when cells were treated for one hour with H_2_O_2_ in addition to etoposide. In contrast, the use of NCS alone or added to etoposide conditions did not induce any change in PMF, an effect also seen with H_2_O_2_ used alone. In conclusion, inducing senescence with X-ray irradiation alone is not sufficient to induce a shift of the PMF. Only the treatment with etoposide and irradiation was able to induce a modification of the PMF similar to what was observed with the UCU. It was also found that this effect was enhanced by the presence of H_2_O_2_ capable of inducing oxidative stress. These results lead to the hypothesis that irradiation and treatment would induce a change in cellular state that could be responsible for the shift of the PMF.

### 3.6. Influence of the Cell Death in the Modulation of PMF

To validate the hypothesis of cell state change, we wanted to know the importance of cell death on PMF. Etoposide is known to induce cell death if used at a high dose [44], so it was important for us to see if the decrease in the PMF was induced by an activation of cell stress or cell death and thus confirm the involvement of cell stress in this phenomenon. For this purpose, FR was analyzed on the first day of treatment (J + 0), the third (J + 3), the sixth (J + 6), and the last day (J + 9) of T24 cells (Figure 6A). Only a significant decrease in the FR of the I/T condition was observed on the last day of treatment, confirming that a long and repeated exposure was required for the loss of the PMF. It was therefore important to know whether after nine days of treatment the significant decrease in FR, and thus PMF, was caused by a significant rate of cell death or by the presence of cell stress independent of apoptosis. To determine the impact of cell death on PMF, a flow cytometry assay with annexin V (apoptosis) and propidium iodide (necrosis) [45] on the T24 cell was performed in parallel with PMF quantification (Figure 6B and Appendix A). An overall increase in cell death was observed for the I/U and I/T conditions compared to U/U cells (Figure 6B and Appendix A). Despite this high level of cell death from J + 0 to J + 6 for the I/T condition than I/U, no change in FR was noticed (Figure 6A). Although there was a significant increase in cell death at J + 3 for the I/T condition compared to the I/U condition, no change in PMF was observed (Figure 6A,B and Appendix A). This suggests that cell death would not be playing a significant enough role to induce a change in PMF at this stage. At J + 9, while there was a significant decrease in PMF for the I/T condition, the level of cell death was greater for the I/U condition than for I/T and U/U (42.17% vs. 22.23% and 4.81%) (Figure 6A,B and Appendix A). If cell death and decreased PMF were correlated, a low FR and high level of cell death should have been found with our I/Ts conditions. These results demonstrate that there is no correlation between cell death and PMF decrease. It can therefore be concluded that the decrease in PMF is observable between the sixth and ninth day of the treatment and that the increase in cell death caused by irradiation with or without treatment is not able to induce a decrease in PMF.

## 4. Discussion

Since the discovery of PMF by the team of Steenkeste et al. [21], no research has been conducted to understand its origin. Therefore, in this present study, we sought to characterize this phenomenon. We have shown that the phenomenon of PMF was not due to morphological and discriminating features of BC cells; all cells in shape and not originating from urine showed PMF. The plasma membrane was then studied and we showed that its permeabilization revealed the crucial role of its integrity in the maintenance of the PMF. Finally, by recreating a strong cellular stress in T24 cells, it was possible to reproduce a PMF similar to the UCUs, suggesting that such a phenomenon was responsible for the loss of the PMF in the UCUs.

Today, early detection of BC is a crucial issue, especially for the increase in the survival rate. BC is one of the most expensive cancers to treat and follow. The average cost of care during the first year after diagnosis varies from USD 47,500 for advanced stage to USD 14,300 for early stage. It can reach over USD 172,000 for a long-term survivor [46]. An early detection of BC would improve the management of the patient, could lower these expenses, and increase the chances of survival. Indeed, the 5-year relative survival is 96% when BC is detected as carcinoma in situ and drops drastically for other stages (70% for non-muscle invasive BCs, 37% for muscle-invasive BCs, and 6% for metastasized BC) [4,47,48].

Thus, research in recent years has focused on new diagnostic methods, based on the search for biomarkers such as free DNA from circulating cells, microRNA, circular RNA, non-coding RNA, proteins, cells, etc. [3,6,7,8]. Despite a continuous search for new biomarkers and the introduction of new FDA-validated diagnostic tests, urine cytology and cystoscopy remain the gold standard tests for BC [49]. The new diagnostics, used in parallel with the reference tests at best, only slightly improve the initial diagnosis and do not reduce the cost. However, the reference tests are not without flaws. Despite its high sensitivity and specificity [50], cystoscopy is particularly troublesome for patients. It is painful, despite the fact that it is performed under local anesthesia, and can lead to complications (e.g., urinary tract infections or pain during urination). These discomforts experienced by patients can lead to loss of follow-up, detrimental to patient survival [51,52]. Conversely, urine cytology, as a non-invasive and low-cost test, is very well experienced by the patient and gives good results for HGBC. Nevertheless, its sensitivity is too low for the diagnosis of LGBC. This is why, lately, research has focused on improving urine cytology, a simple and low-cost technique. Since the identification of PMF by the team of Steenkeste et al. [21], new possibilities for its improvement have emerged.

Based on their research, we wanted to understand and characterize the PMF. From this perspective, it was essential to know if the PMF was a phenomenon induced by the morphological feature, called the eclipse effect, or if it was a biological event induced by the staining. We were interested in the morphological feature because a large number of studies describe the large size of the cell nucleus (greater than 70% of the cytoplasm volume) as a key factor in the identification of a UTC [35,53]. After labeling the nucleus and cytoplasm, we showed that the cell nucleus was large but not large enough to induce this phenomenon. We were therefore able to deduce that this was a biological feature, which occurs during staining.

Based on the assumption that only UTCs show PMF, we searched for cell lines not expressing PMF to compare their cell composition with T24 cells. After testing many cell lines of various origins, healthy or tumor, we found that all cells showed PMF after Papanicolaou cytology staining.

In order to understand this phenomenon, it would have been useful to look at the dyes used, the Sakura^®^ film that covers the cytology, and the glue used for mounting. The results are not described in this publication but have been researched in parallel to this study. To determine the role of the dyes in PMF, a study was conducted on the different staining steps where one of the dyes (hematoxylin, EA50, or OG6) was removed independently from the others to understand their respective importance. According to this study, EA50 and OG6 are essential to visualize the PMF and hematoxylin and EA50 for the intensity of the PMF (data not shown).

As we could not characterize the chemical origin of the PMF nor find a cell line without PMF, we decided to set up a software able to give the precise PMF of a cell in order to determine if the cells already tested showed a variation of PMF or not, depending on the origin of the cell.

This software, Perifluo QCG, offered us the necessary elements to go from a qualitative analysis where we were looking for a cell model without PMF to compare with a model with PMF, to a quantitative analysis where we could significantly determine a variation in PMF. As a result, our software can quantify every cell on the cytology extremely quickly, giving us a great amount of information. Perifluo QCG uses a simple process whereby it calculates the cytoplasmic and membrane fluorescence (up to 5 pixels around the membrane), while deducting the background. Once the data is obtained, the PMF value is calculated by operating FR.

If this FR value is greater than 1, the membrane fluorescence is more important and a PMF is therefore present; conversely, if the FR value is less than 1.

Thanks to this tool, the determination of the FR of cells already tested was facilitated, allowing us to easily distinguish the variations of PMF according to the cell lines used. The use of the Perifluo QCG software did not reveal any significant differences in PMF and confirmed that all cells, after Papanicolaou staining, constitutively express this fluorescence. The software also allowed us to confirm the hypothesis that UCUs would have an FR lower than 1 and therefore would not express PMF, unlike UCs recovered from the UroThelium (UCUT) which have an FR higher than 1. From these results, it is clear that the transition from the bladder to the urine induces a change in the cell that can interfere with the staining. Thus, we hypothesized that when UCs desquamate and find themselves in an environment unfavorable to their survival, various cellular mechanisms intervene and induce a strong cellular stress [24,25], leading to a membrane alteration responsible for the PMF decrease. To confirm the importance of the plasma membrane in the PMF phenomenon, we used a detergent to permeabilize the plasma membrane of our cells. Initially, a decrease in PMF, similar to that obtained with UCUs, was observed. This decrease was also observed after induction of intense stress, independently of the activation of senescence. These results confirm that the membrane plays an important role in PMF and that PMF is sensitive to strong cellular stress.

In view of these results, we suggest that the plasma membrane plays a primary role in the PMF, especially regarding its composition. We hypothesize that one of the main stresses on UCs could be induced by anoikis after UCs detachment. Anoikis induces the entry of the UCs into apoptosis, which leads to a change in the composition of the plasma membrane [38,54]. The resistance of tumor cells to this phenomenon and its importance in metastatic dissemination have been widely described [55,56]. Therefore, we suggest that UTCs, like many tumor cells, are able to resist anoikis when suspended in urine. This resistance prevents the induction of cellular stresses and thus the alteration of the plasma membrane.

Our study shows that the plasma membrane plays a major role in PMF; however, this criterion alone would not be suitable to determine the presence or absence of UTCs (as, for example, a common live/dead cell staining) since not all live cells are necessarily UTCs.

To confirm this hypothesis, it would be interesting to study the activation of anoikis on our T24 cells using a drug, as proposed by Terasaki et al., whose research showed that norcantharidin induces anoikis in their cell model [57]. A second possible approach would be to directly induce a change in the composition of the plasma membrane by inhibiting the production of phosphatidylcholine (PC), a major phospholipid of the plasma membrane (representing 40% of the phospholipids) [58]. The preliminary study which was conducted shows that inhibition of phosphatidylcholine biosynthesis by miltefosine for 24 h induced a significant decrease in PMF independent of apoptosis (Appendix A). Looking further, it would be useful to assay PC to determine its expression level in the membrane and highlight its importance in PMF.

## 5. Conclusions

In summary, we propose that PMF is not only observed in UTCs but in any cell in good shape. We speculate that the high resistance of UTCs to external aggression allows them to survive better in urine, unlike UCs, which, once in suspension, die more easily and lose their PMF.

## Figures and Tables

**Figure 1 cancers-14-02171-f001:**
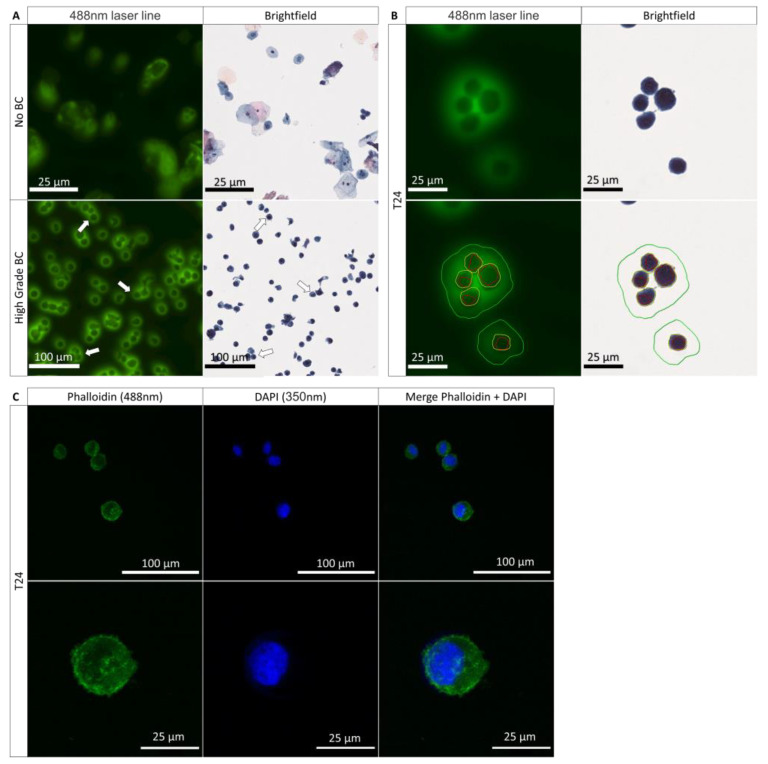
Characteristics of the fluorescence from healthy and tumoral urothelial cells. (**A**) Urinary cytology acquired on the scanner of a healthy patient (no BC) and a patient with an HGBC (high-grade BC) in bright field light and at 488 nm; arrows show the tumor cells expressing a PMF. (**B**) Scanning images of T24 cells from a cell culture with a demarcation of the nuclear membrane in red, the plasma membrane in yellow, and the PMF, in green in bright field light and at 488 nm. (**C**) Immunofluorescence labeling of the cytoplasm with phalloidin (488 nm) and of the cell nucleus with DAPI (350 nm) to determine the presence of an eclipse effect.

**Figure 2 cancers-14-02171-f002:**
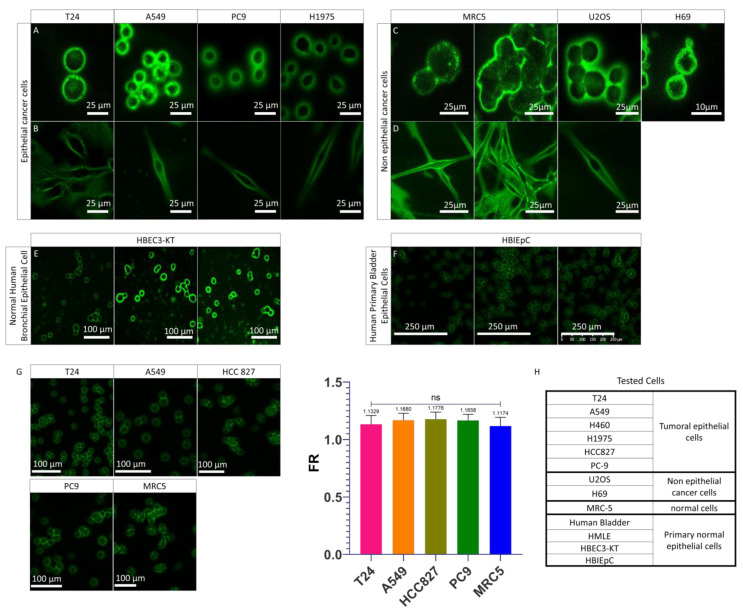
PMF on different cells line. (**A**) Images acquired on confocal microscope of epithelial cancer cell lines in suspension mounted on the slide as a urinary cytology. (**B**) Z-stack images ac-quired on confocal microscope of adherent epithelial cancer cell lines mounted on slide as urinary cytology. (**C**) Images acquired on confocal microscope of non-epithelial cell lines in suspensions mounted on slide as urinary cytology. (**D**) Z-stack images acquired on confocal microscope of adherent non-epithelial cell lines mounted on slide as urinary cytology. (**E**) Images acquired on the scanner of a normal human bronchial epithelial cell line. (**F**) Images acquired on the scanner of a normal human primary bladder epithelial cell line. (**G**) Images acquired on the scanner on previously tested cell lines and quantification of their PMF with FR. Bar graphs represent the mean (SD) (ns > 0.05). (**H**) Origins of tested cells showing PMF.

**Figure 3 cancers-14-02171-f003:**
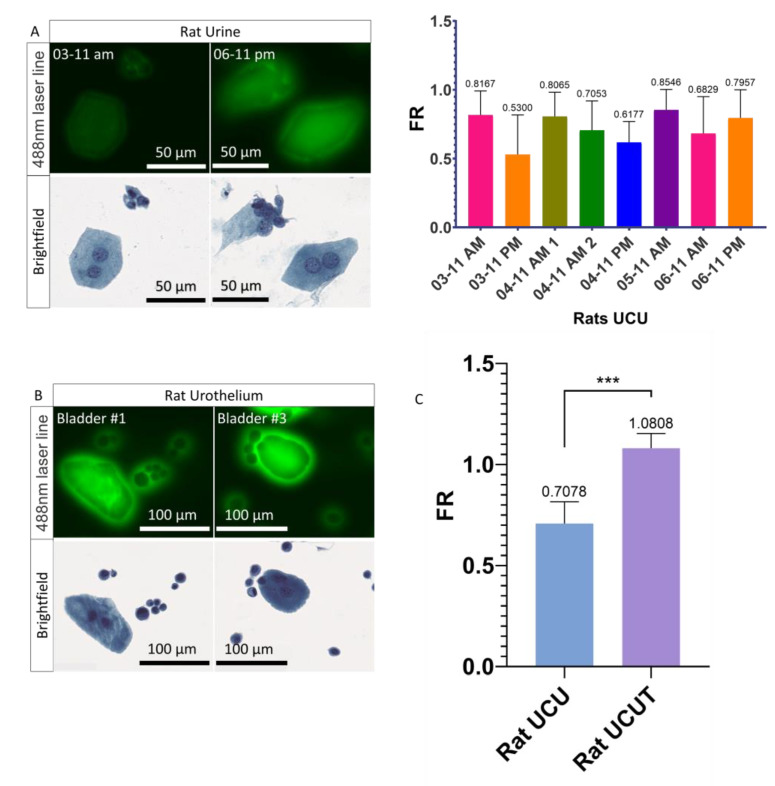
Fluorescence and PMF of UCUs and UCUTs. (**A**) Images acquired on the scanner of UCUs and quantification of their PMF. Three to 5 rats were pooled by sample. (**B**) Images acquired on the scanner of UCUTs. (**C**) Quantification of the PMF of UCUs versus UCUTs. Bar graphs represent the mean (SD) (*** *p* <0.001).

**Figure 4 cancers-14-02171-f004:**
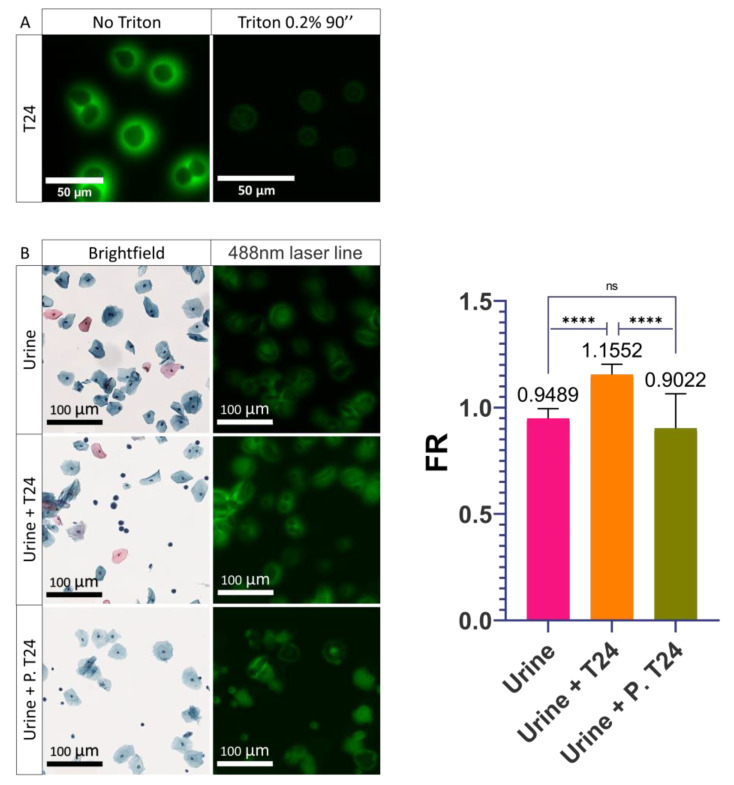
Influence of the permeabilization of the plasma membrane on cells. (**A**) Images acquired on confocal microscope of the permeabilization of T24 cells with 0.2% Triton for 90 seconds. (**B**) Images acquired on the scanner of healthy urine alone (Urine), supplemented with T24 cells (Urine + T24), or supplemented with T24 cells permeabilized (Urine + P. T24) with 0.2% Triton for 90 seconds and quantification of their PMF. Bar graphs represent the mean (SD) (ns > 0.05; **** *p* < 0.0001).

**Figure 5 cancers-14-02171-f005:**
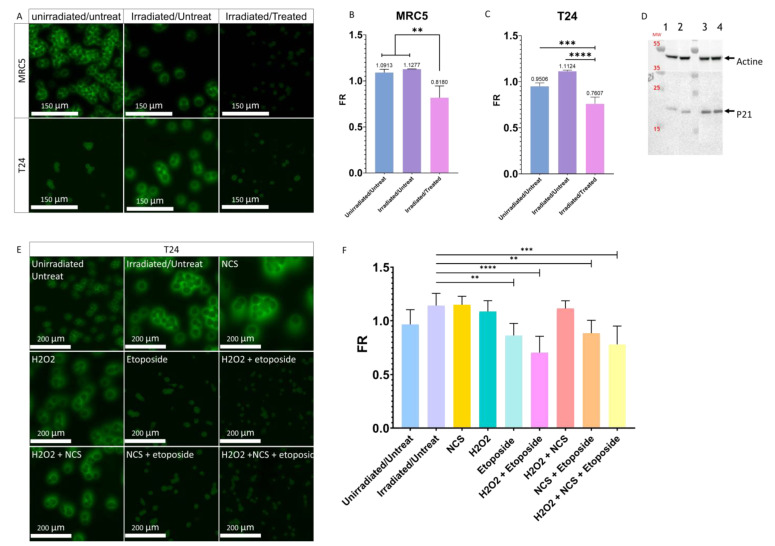
Involvement of cellular stress in the modulation of PMF. (**A**) Effect of irradiation with or without treatment on the PMF of MRC-5hTERT and T24 cells after nine days. (**B**) Quantitative determination of the effects of irradiation with or without treatment on the PMF of MRC-5 hTERT after nine days. Bar graphs represent the mean (SD) (** *p* < 0.01). (**C**) Quantitative determination of the effects of irradiation with or without treatment on the PMF of T24 cells after nine days. Bar graphs represent the mean (SD) (*** *p* < 0.001); **** *p* < 0.0001). (**D**) Western blot of P21; 1: Unirradiated/Untreated T24 cells grown with serum, 2: Unirradiated/Untreated T24 cells grown without serum, 3: Irradiated/Untreated T24 cells grown without serum, 4: Irradiated/Treated T24 cells grown without serum. The original blots could be found in the Appendix A. (**E**) Effect of the different treatments and molecules to induce senescence and cellular stress on the T24 cell PMF after nine days. (**F**) Quantitative determination of the impact of different treatments and molecules to induce senescence and cellular stress on T24 cell PMF after nine days. Bar graphs represent the mean (SD) (** *p* < 0.01; *** *p* < 0.001); **** *p* < 0.0001).

**Figure 6 cancers-14-02171-f006:**
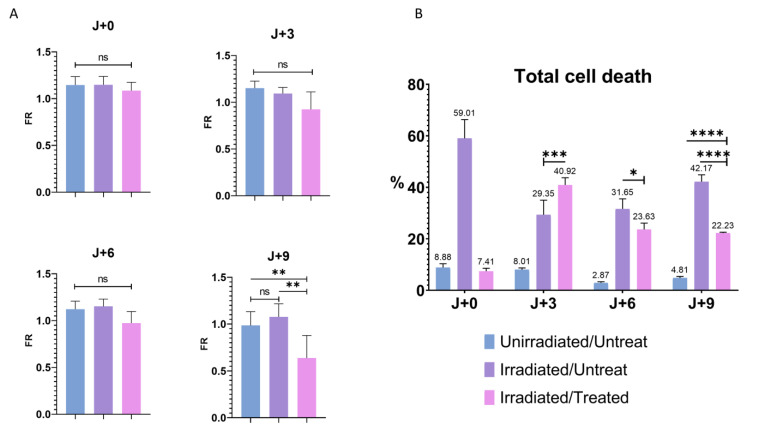
Cell death and PMF. (**A**) Quantitative determination of PMF after irradiation- and treatment-induced senescence on T24 cells at J + 0, J + 3, J + 6, and J + 9. Bar graphs represent the mean (SD) (ns > 0.05; ** *p* < 0.01). (**B**) Flow cytometry determination of early apoptosis, late apoptosis and necrosis combined after induction of cellular stress by irradiation and treatment on T24 cells, labeled with annexin V and propidium iodide. Bar graphs represent the mean (SD) (* *p* < 0.05; *** *p* < 0.001); **** *p* < 0.0001).

## Data Availability

The data presented in this study are available on request from the corresponding author.

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
