# Peer review of "Characterization of the Peri-Membrane Fluorescence Phenomenon Allowing the Detection of Urothelial Tumor Cells in Urine"

_cancers, 2022, doi:10.3390/cancers14092171_

Round 1

Reviewer 1 Report

Charly Gutierrez and co-authors have been studied fluorescent properties of plasma membranes of urothelial tumor cells found in urine cytology. The impact of research is significant and allow patients to be diagnosed at early cancer stage.

Line 53: there is no explaination of T2 and T4 classification

Line 83: "Some diagnostic tests also require expensive equipment". What kind of diagnostic method?

Line 144 and 149: you mentioned 10% FBS twice

Figure 2 and 3: There is no [r.u.] units of Y axis.

Reviewer 2 Report

This manuscript studies the phenomenon of perimembrane fluorescence as it pertains to the potential detection of urothelial tumour cells in urine. The authors have included a substantial amount of experimental detail well-summarised in the figures and discussed in suitable detail in the text. Some key findings regarding changes in perimembrane fluorescence have been detailed and a sensible approach to quantification of perimembrane fluorescence has been presented. The manuscript is clearly written and provides suitable context for the research that clearly outlines the novelty and impact of the findings. There are some points which the authors can address and these are detailed below.

  1. Page 3 lines 111-113 “Thus, our results . . . to the healthy cell.” I don’t think that the authors have provided evidence that the PMF is directly caused by the survivability of the tumour cells. Whilst it is obvious that if the tumour cells can remain viable under the stressful environment, then they would be expected to retain at least some of their PMF, but this is not the same as saying the PMF is caused by the resilience of the tumour cells.
  2. Figure 2H. Change “No epithelial cancer cells” to “Non-epithelial cancer cells”.
  3. Page 10 line 395 “. . . ranging from 0.6177 . . . for the maximum) . . .” It seems that these FR values may have been given with too many significant figures. Please ensure that the number of significant figures used in these FR values are consistent with the experimental errors involved in their computation. This applies here and throughout the manuscript where FR figures are given.
  4. Page 12 lines 465-466 “Western blot analysis . . . (Figure 5d) . . .” As written, this statement is incorrect and contradicts your later statement at line 469. The I/T condition also expresses more P21 than the U/U condition. Please reword this statement accordingly.
  5. Page 14 lines 516-517 “. . . it is found . . . 4.81% respectively) . . .” The order of the percentage figures is incorrect and confusing. Here you are comparing the I/U condition to the I/T and N/N conditions, so the percentages should read: “. . . (42.71% vs. 22.23% and 4.81% respectively) . . .” Also, what does the “N/N” signify? I have not seen it used earlier in the manuscript and have assumed that you meant U/U. Please correct and clarify.

Reviewer 3 Report

The manuscript by Rémy Pedeux, et. al. described the evaluation of a newly developed diagnosis assay, so called perimembrane fluorescence (PMF) for urothelial tumor cells in urine. A software was created to quantify PMF from both normal and tumor cells. The authors found that plasma membrane integrity is important for the maintenance of PMF. Tumor cells are more resistant to environmental stress, which might be the reason for tumor cells to maintain membrane integrity and PMF. This study indeed offers useful information for the readers in the related fields. However, some issues have to be addressed before further consideration.

  1. It‘s well known that the signal we can get relies heavily on which position is focused when applying laser scanning confocal microcopy to take cell images. False result is easily to be introduced when comparing the brightness during imaging. I think a general and normal fluorescence microscope will be better for PMF determination and quantification. The authors have to show the readers some evidences.
  2. Since membrane integrity is important for PMF, is it possible to diagnose UTC using a common live/dead cell staining assay? Explain why?
  3. The authors have to strengthen and highlight their contribution in the abstract and conclusion part, in comparison with previous studies.
  4. The paper written is hard to understand, the authors have to reorganize the language extensively.

Round 2

Reviewer 3 Report

No more comments.